# Predicting Player Churn of a Free-to-Play Mobile Video Game Using Supervised Machine Learning

**Kuzma Mustač †, Krešimir Bačić †, Lea Skorin-Kapov and Mirko Sužnjević ***

Faculty of Electrical Engineering and Computing, University of Zagreb, Unska 3, 10000 Zagreb, Croatia;
kuzma.mustac@fer.hr (K.M.); kresimir.bacic@fer.hr (K.B.); lea.skorin-kapov@fer.hr (L.S.-K.)
* Correspondence: mirko.suznjevic@fer.hr
† These authors contributed equally to this work.

**Abstract:** Free-to-play mobile games monetize players through different business models, with higher player engagement leading to revenue increases. Consequently, the foremost goal of game designers and developers is to keep their audience engaged with the game for as long as possible. Studying and modeling player churn is, therefore, of the highest importance for game providers in this genre. This paper presents machine learning-based models for predicting player churn in a free-to-play mobile game. The dataset on which the research is based is collected in cooperation with a European game developer and comprises over four years of player records of a game belonging to the multiple-choice storytelling genre. Our initial analysis shows that user churn is a very significant problem, with a large portion of the players engaging with the game only briefly, thus presenting a potentially huge revenue loss. Presented models for churn prediction are trained based on varying learning periods (1–7 days) to encompass both very short-term players and longer-term players. Further, the predicted churn periods vary from 1–7 days. Obtained results show accuracies varying from 66% to 95%, depending on the considered periods.

**Keywords:** player churn; free-to-play; player behavior analysis; mobile game; machine learning

## 1. Introduction

Game design and development have come a long way since their humble beginnings, from very simple 2D platformers, 3D games of various genres, to today's realistic, breath-taking graphics and immersive virtual reality games. It is difficult to determine which gaming genre or platform is the most popular, as player preferences and demands vary and change quickly. Developers need to adapt as the active playerbase determines the success of a game. With the evolution of the Internet, games have become easily accessible and advertisable, the popularity of multiplayer games has skyrocketed, and countless players now share their feedback over forums and other platforms.

Similar to the evolution of the games themselves, monetization tactics have evolved accordingly. Before the Internet was stable and fast enough, most games had a set price—once a customer bought a game, there was no need for additional purchases. In some less frequent cases, developers released additional downloadable content (DLC), which could extend a game's features for a price, usually less than the cost of the game. Later, the massively multiplayer online role-playing game (MMORPG) genre took a different approach—monthly subscriptions. The most notable example is *World of Warcraft* (WoW), in which the subscription model worked due to the game's unrivalled popularity (WoW reached 12 million subscribers in 2010 [1]). Less-popular MMORPGs could not afford to use such a model, as players were often unwilling to pay monthly subscription fees. Consequently, game providers would offer the game for free, while offering players the option to purchase cosmetic items or some kind of boost which would give them an advantage over non-paying players [2]. This business model has grown to be widely

regarded as the F2P (free-to-play; also referred to as freemium [3]) model, and has proven to be very successful in a wide range of PC, console, and mobile games.

Studies have shown that if a F2P game offers in-game purchases, the vast majority of players tends to spend no money, or a very small amount, on virtual goods inside the game [4]. According to the 2019 Swrve Monetization Report, providing an in-depth study of player spending habits in freemium mobile games, only 1.6% of players make an in-app purchase, while 72% of paying players make only one purchase [5]. Consequently, a very large portion of the profit is generated by a very small percentage of players, which have been referred to in the past as "whales", with studies having found the majority of such players to fit to the gamer stereotype of being young males [6]. The term "whales" was originally used in a negative way, as developers used it to describe customers who are so wealthy, they could sell anything to them. The term is still used in the gaming community, albeit not in a negative manner, but rather to refer to players that simply spend a lot of money on the game [7]. It has further been argued that rather than such players being spenders which can afford anything offered to them, mostly they are simply unaware of the actual amount they have spent on the game, likely due to the large number of low-value transactions [8].

Given player spending habits and the monetization potential of the F2P model, it is clearly important for game publishers to identify and create content that will appeal to paying customers, so as to keep the players playing the game and maximize achieved revenue. The other relevant revenue stream for developers is marketing—players are shown commercials, commonly for other games from the same developer, or from other developers. For this business model, it is also imperative to keep the players engaged as much as possible, so that more commercials are shown to them, thereby yielding increased revenue from this model. The question arises as to what implications marketing decisions have on player spending habits and potential churn, calling for the need to micromanage the combination of these two revenue streams. In the scope of the study conducted in this paper involving a F2P mobile game, we received feedback from the game developer that their records show that if a paying player is targeted with ads, they tend to decrease their spending amount, and are more prone to leaving the game. Nevertheless, no systematic evidence of such an effect of advertising on a user's retention was reported in a previous study on the topic [9]. In general, the more active the player, and the longer the player is in the game, the greater the chance the player will become a paying customer through completing a purchase with real money or by creating revenue through viewing ads. As a consequence, it is imperative for game designers and developers to predict which of their customers will leave the game and when, so as to optimize marketing decisions and potentially to increase overall retention and revenue.

The aim of this paper is to investigate the potential of applying machine learning (ML) based models for predicting player behavior in terms of staying active in the game or leaving the game, based on features derived from previously collected gameplay event logs. ML models can provide a fast and efficient way of analyzing player behavior in such a dynamic and varying environment, and can also help developers see certain trends in their player base that would not have otherwise been noticed. The other approach is using analytical models created by experts. These expert models can be very expensive to create from the standpoint of both the expertise of people creating them, as well as from the standpoint of time. While such an approach might yield comparable results, the created models need to be adapted to specific game changes, the lifetime cycle of the game, etc., which means spending a lot of expert man-hours on repetitive tasks. Moreover, what is problematic about using expert approaches in the area of F2P games is the sheer volume of data (e.g., the game addressed in this study has had, at one point, around 200,000 daily new players). Therefore, ML algorithms fueled by fully automatic data gathering and processing functions provide a reasonable choice for study and application.

From a game-provider's perspective, the developed models may have different purposes, such as to correctly identify players which will soon quit the game so that they can

either be targeted with advertisements and consequently monetize them, to offer them a limited number of specific offers and discounts to keep them active in the game, or any other strategy depending on the business model. For different goals, the developed models need to have different properties.

As a case study, we analyze a large dataset of anonymized player data provided by a European game developer, corresponding to a free-to-play interactive storytelling game available on the international market. The game can be categorized as a casual mobile game (or even as a hyper-casual mobile game) which makes it a useful case study, as casual games are one of the most popular in today's mobile gaming market [10]. In the game, the player can read stories comprised of chapters. To access a chapter, the player first needs to pay a type of in-game currency, which regenerates over time. If the player wants to read at a higher pace, then he/she needs to purchase in-game currency with real money. While reading each chapter, the player is offered a set of choices he/she can make, and certain choices require a second type of in-game currency which can only be obtained with real money. Additionally, the player can customize the appearance of his/her character in-game though purchasing clothing items with this second type of in-game currency. Interactive storytelling games have a huge market potential, and the game under study in this paper has spent multiple weeks in the top 100 games on the iOS app store in the USA.

The available dataset comprises logs corresponding to over 80 million players, and was collected over a time frame of approximately four years (January 2018 to January 2022). The dataset was used to construct smaller datasets on which the models are trained and validated, as performing analyses on the entire dataset is very computationally intensive and the long-term trends might skew the models. Each particular dataset was created by determining a particular time frame. Different time frames were taken to ensure that a certain point in the game's lifecycle does not have a specific impact. We did specifically discard the Christmas period, because of special offers and the significantly different behavior of the players in that period. The models were aimed at predicting whether or not a given player will continue playing or leave the game in a certain future time frame.

The paper is organized as follows. In Section 2, we give a brief overview of related work concerned with analyzing and predicting player churn. Our problem description, definition of relevant parameters, and initial churn analysis are given in Section 3. Section 4 outlines our research methodology in terms of dataset collection, processing, and churn-prediction model development. Detailed classifier results for proposed churn-prediction models are given in Section 5 for various feature extraction and churn periods. Section 6 provides a summary of key findings and provides an outlook for future work.

## 2. Related Work

The topic of analyzing player behavior and its application in business intelligence has evolved over the years, as have the games' business models. During the prevalence of the pay-to-play (P2P) model, and especially the subscription-based model that was popular amongst MMORPGs, *player churn rate* (i.e., a measure indicating how many players stop playing a game over a certain time period) and *player retention rate* (i.e., a measure indicating how many players continued playing a game over a certain time period) were the most important metrics investigated. In [11], the authors measured player churn through the GameSpy service, and identified that it follows a power-law distribution. Similar results were confirmed in [12], where findings indicate that an average player's interest in playing these games evolves according to a non-homogeneous Poisson process, whose intensity function is given by a power law, with the Weibull distribution being the most appropriate to describe playing time per player. Churn in the MMORPG *Everquest II* was analyzed in [13], with a focus on social networks and their impact on churn. The authors present churn-prediction models, but their precision and recall are relatively low—around 50%. The dataset from *Everquest II* was also used to predict churn using supervised and unsupervised learning methods in [14], achieving precision and recall up to 75%. The same

authors extended their previous work in [15] by closely looking at player behavior and formulating four types of churner behavior and four types of active player behavior. Their results indicate top accuracy results of 92.9%, whilst most of the models have values of performance metrics around 70%. Some researchers have also investigated the effects of network parameters (i.e., latency and jitter) on player departure in MMORPGs [16]. It should be noted that the examined player departure does not refer to players leaving the game forever, but rather players ending the specific session. The authors created a model which can assess when the players will end the session based on network quality—high delay and high packet loss result in very short sessions. The results show that delay jitters are less tolerable than absolute delays and that packet loss is more tolerable than packet delay. Churn and retention prediction was conducted for other major AAA games, such as *World of Warcraft* [17], *Just Cause 2* [18], and *Destiny* [19], but also for popular F2P games such as *Top Eleven—Be A Football Manager* [20], as the F2P model increased in popularity.

The generation of revenue in F2P games is a highly discussed topic among game developers and publishers, given that these processes are much more complex than in P2P models and are driven by advertising and in-game purchases. In [21], the authors focus on predicting when a player will leave the game, with the goal being to open unique opportunities for companies to increase revenue contribution and provide incentives for players to stay or to initiate a new lifetime in another game. The chosen game genre was casual social games. Three core terms for solving the problem are high-value players, activity, and churn. To be classified as high-value, a player needs to be in the top 10% of players in terms of generated revenue. An active high-value player is considered to be a high-value player who has played the game at least once in the past two weeks. In this paper, the churn of an active high-value player corresponds to the player being inactive during a specific period. The churn prediction was modeled as a binary classification problem, classifying players according to the categories *churn* or *no churn*. After the classification, the high-value churning players were divided into three different test groups. In two of the groups, the players were already churning, and one of the groups received a free amount of in-game currency, while the other did not. The last group was composed of players who were still active, but who were showing signs of churning. They also received a certain amount of free in-game currency. The results showed that providing a substantial amount of free in-game currency to high-value players affected by churning did not significantly affect the churn rate.

Different genres of F2P mobile games have been studied in terms of player churn—role-playing games [22], puzzle games [23], social games [20], etc. It is challenging to generalize research findings, considering that many games have unique actions that a user can perform, and the habits of players often differ across different games. Studies that try to generalize findings have been performed across multiple games, such as [24], in which the authors create classifiers for churning players and remaining players on a dataset comprising five different games, and using only non-game-specific features (e.g., time between sessions). Different types of mobile games have also been combined in certain studies for the evaluation of player churn. In [25], the authors created a dataset from a player simulation game, a racing game, and a music game to predict player disengagement (i.e., players spending less time in the game or logging in less frequently) in those games using specific features related to game and event frequency features, which can be generalized over different games. The results indicated that biased datasets can easily lead to false classifiers, and that disengagement over varying dates can yield better performance over other approaches. Disengagement alongside churn has been a topic of research in [26], where the authors also looked at the freemium strategy game and evaluated different machine learning algorithms, concluding that the random forest model is the best for predicting churn when implementing the "two windows" approach (i.e., defining time windows in which players do and do not appear to be active).

Different approaches towards churn prediction and towards metrics which describe churn have been presented in two subsequent studies by Periáñez et al. [27,28]. The authors

use survival ensembles (a group of survival tree models similar to the random forest approach). Whilst first working with binary classification [27], in the second paper [28], the authors extend the models by predicting the level obtained in the game and the playtime. While the dataset is large, the specific prediction analysis has been conducted on thousands of players—primarily whales—while the short-term players who quit the game quickly were not addressed.

Another approach for predicting player churn [29] is using a recency, frequency, and monetary value framework which takes into account features related to user lifetime, intensity, and rewards (RFM-LIR model). The authors define different churn types based on the windowed approach, starting absence, engagement related to average engagement, and switching behavior (i.e., decreased trend in engagement). AUC values are between 0.53 and 0.71 for developed models.

The research in this paper differs from previous work, as it focuses on a highly dynamic game in terms of player acquisition and player churn. This paper assesses a game in which approximately 60% of the players leave the game after only one day of gameplay, and the influx of new players in the game is counted in the tens of thousands per day. Therefore, because developers have a very short time to react to players leaving their game with any in-game action, we focus on very short time frames for the machine learning algorithms—up to one day—and the results of this research show that developer reactions are needed on even shorter time frames (i.e., several hours), which will be addressed in our future work. With a high churn percentage observed after a single day, there is a need for rapid prediction models, such as that presented in [23], while the majority of related work focuses on much longer time frames (e.g., weeks or months). In addition, our analysis has been conducted on samples of hundreds of thousands of users, as opposed to samples of thousands of users prevalent in the related work. We note that, during the course of our data collection, the influx of new players gradually decreased from over 200,000 players daily to only 20,000. Therefore, we focus on a specific problem which is addressed on several time scales and on a dataset comprising millions of players and partitioned in a way that every specific research question is evaluated on hundreds of thousands of players.

### 3. Problem Description

The data on which the research was conducted was provided by an internationally established European game developer. The game is an interactive, story-driven game in which players click through animated stories, freely choosing any of the given dialogue options. The stories are divided into chapters, which require tickets to unlock. Spent tickets can be regenerated over time or can be bought using premium in-game currency. The premium currency can also be spent on customizing the players' avatar, or to unlock premium dialogue options in various chapters and stories.

We once again highlight that our main focus is on predicting player churn and player retention. While churn is commonly linked to players leaving the game due to lack of motivation to continue playing, player retention refers to a set of decisions which impact the game and consequentially player behavior, aiming to motivate them to keep playing the game for a longer period of time (retaining the players attention). This can be accomplished in a number of ways, i.e., by handing out free premium currency, providing special offers and bonuses, optimizing other game systems, etc. Retaining the players' motivation keeps the playerbase healthy, improves the overall game experience, and increases overall profit for the game developers.

#### 3.1. Definition of Churn Parameters

The time constraints are defined as different time periods used in the process of tracking player behavior. Under the term player behavior, we consider the following features of player behavior, similarly to [25,29]: (1) frequency features (i.e., how often events occur), (2) monetary features (i.e., lifetime value of the player), (3) lifetime and intensity features (i.e., what percentage of overall time is spent in the game at which point

in a player's lifetime), and (4) game-specific features (e.g., chapters completed). As shown in Figure 1, player behavior tracking starts as soon as the player installs the game (the *firstInstallTime* attribute). The feature extraction period lasts one, two, three, or seven days and is followed by the churn period, in which player churn is monitored and lasts between one and seven days. If the system notes any player activity in the churn period, the player is considered a non-churner. On the other hand, if no player activity is noted in the churn period, then the player is assumed to be a churner. The ground truth period serves as a cut-off time for monitoring player churn and lasts fourteen days after the feature extraction period has ended. This period serves as a validation period to check whether players which would be labeled as churners depending on the defined churn period will or will not return to the game.

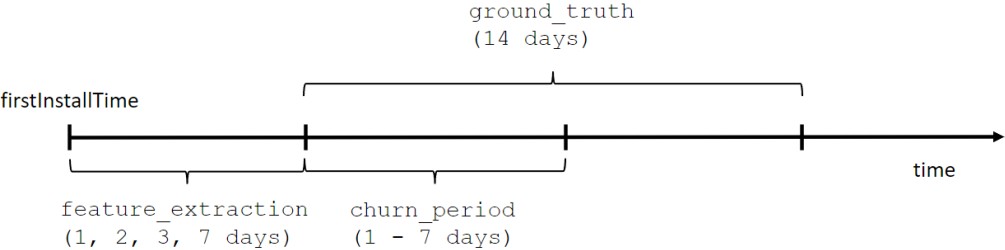

**Figure 1.** Time constraint definitions.

The aforementioned churn period is the period in which the users' activity is monitored and a prediction is made as to whether the player will quit the game or continue playing. Another way of looking at this is by tying a churn label to every player (the players' user profiles, to be exact), which serves the same purpose. If the user first installed the game on a certain date, and on the very next day, the system has not logged any activity from the same user, then the user is estimated to have churned and the churn label would be positive. Alternatively, if any activity is detected on the next day following installment, the churn label would be set to negative—the player has not churned. The churn label is, by definition, an estimate, and it is useful to calculate how accurate that estimate is with respect to the ground truth definition of a certain player's churn.

To showcase how these periods are related and how accurate churn prediction is based on only one day, we plot values extracted from our dataset during a random time frame (Figure 2). The graph presents the period from 27 April 2020 to 24 May 2020. The graph portrays how many new users first installed the game on a certain date (numOfUsers), how many of those users were estimated to churn by not showing the next day (*churnEstimates*), and how many of the estimated churners were "real" churners (*churnReals*, according to the ground truth period definition of 14 days). Calculating the average accuracy of the churn estimates over the given period yields an estimate accuracy of 83.65% using just the one-day estimation.

If we were to observe the user for one more day (i.e., two days after having installed the game), the accuracy of the estimation will rise to 90.09%. The graph shown in Figure 3 shows how the estimation accuracy rises with an increase in observation duration. If the user has not shown up in the game (i.e., no activity detected) for seven days, there is a 98% chance that they will not show up in the 14-day ground truth validation period.

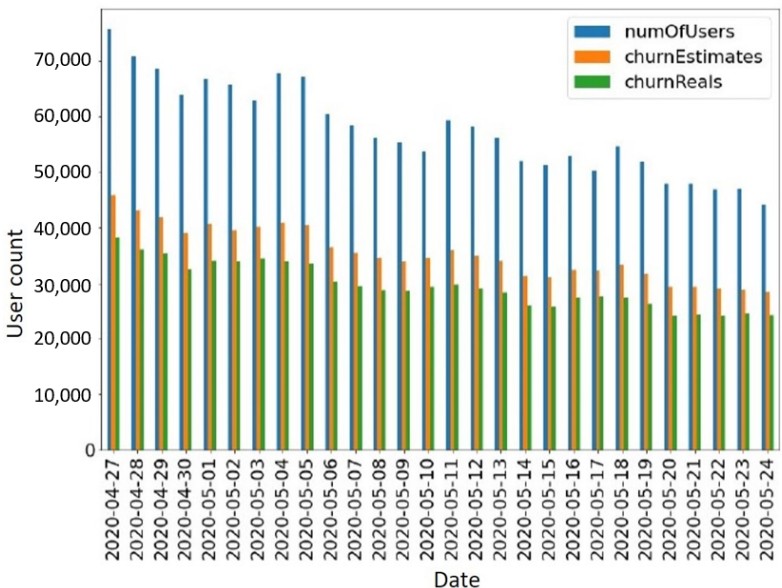

**Figure 2.** Illustration of player churn estimates based on player activity on the day following game installment as compared to ground truth data considering player activity over a 14-day period.

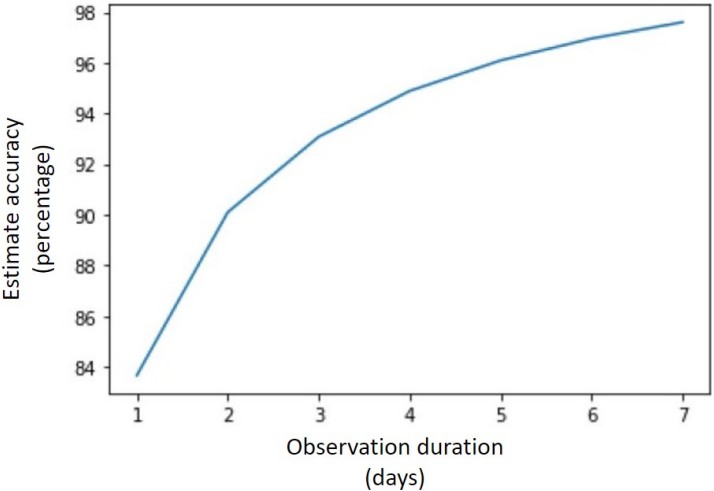

**Figure 3.** Estimate accuracy in relation to observation duration.

### 3.2. Initial Churn Analysis

Before continuing research on ways to combat player churn, a solid case must be made that the churning problem even exists. We conducted a short statistical study on a sample of our dataset (starting from 27 April 2020 until 24 May 2020, the sample is later referred to as dataset D3). From this pool of players, we plot the activity distribution in Figure 4. To clarify, for each of the new players in the dataset, we note whether he/she has been detected in-game within 7 days of the initial day he/she installed the game (day 0 denotes the day the user has installed the game). Considering this sample, we found that, out of the 1.5 million players considered, 61.81% of them did not show in the game after the first 24 h, as shown in Figure 4. After only three days, that number rose to 81.9%. This clearly shows that player churn is a real concern, and the costs and efforts associated with finding ways to combat player churn are justified.

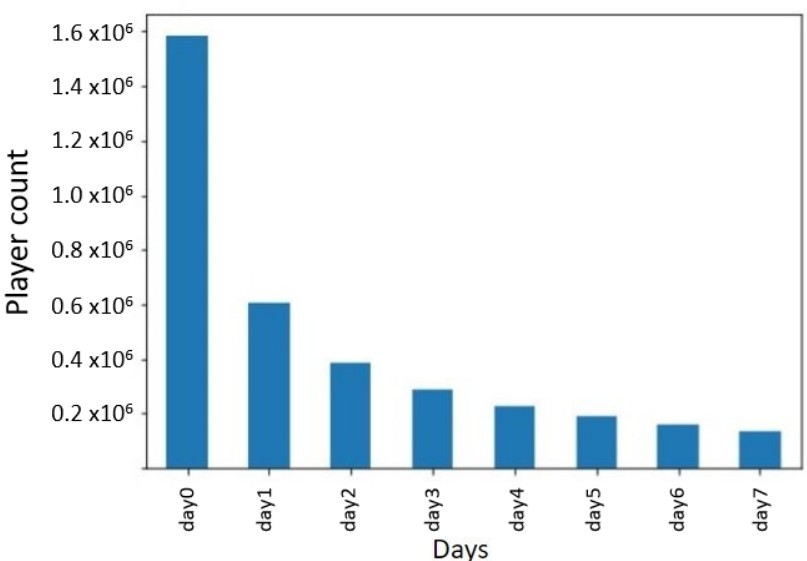

**Figure 4.** Number of players active in the game on a certain day after having installed the game.

We additionally perform a separate analysis to ascertain player churn distribution by considering player activity rather than absolute time frame. In our concrete case of an interactive storytelling game, we consider the number of chapters a player completed reading in a given story prior to churning. The results were unsurprising: a disproportionate percentage of players churned during one of the early chapters, with the first two chapters often exhibiting user retention percentages below 40%. In contrast, players that reached later chapters were significantly more likely to finish them, with completion percentages varying between 48% and 49%, with respect to the total number of players for each chapter. In terms of a single (first) story, more than 75% of the playerbase dropped out before chapter 10, as presented in Figure 5. The analysis presented in Figure 5 was conducted on all players arriving in the game in the period of 4 weeks, from 26 October until 22 November 2020.

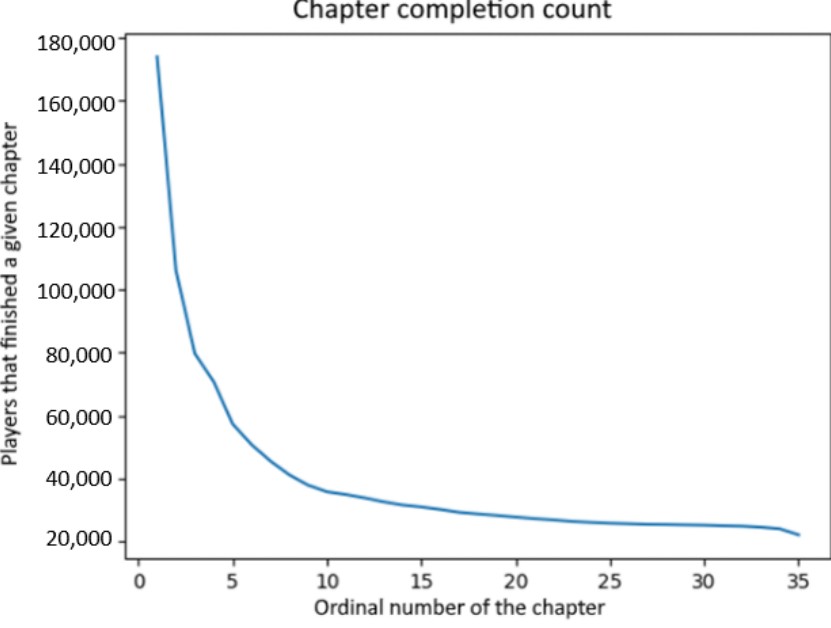

**Figure 5.** Player churn distribution by story.

Based on the demonstrated results, we inferred that early churn detection plays a crucial role in its long-term prevention. The insight provided by preliminary analysis led

us to conclude that greater focus should be placed on the first several days of a user's playtime, as it could prove to be highly indicative of their willingness to continue playing the game for an extended period. Moreover, focusing on this early period would enable the game provider to engage a large portion of the audience that would stay in the game for a very brief period.

## 4. Research Methodology

This section describes the various aspects of our research methodology employed during the study. The research methodology flowchart is shown in Figure 6. The data was collected by the development team using an event tracker that logs all player events during game play and then parsed and imported into a database. It was then analyzed for potential problems (e.g., missing events labeling the start and end of the session, incorrect time stamps of events, wrong information about the player's life-time values), and any notable issues found were dealt with through the data-preparation step. Cleaned data was used for building player profiles, which were, in turn, used as the input for training various machine learning algorithms. The algorithms were compared using established metrics and the model which exhibited the best performance was selected. The data used in the calculation of the profiles' features was bounded to certain predetermined time intervals—the feature extraction and churn periods. Further research focused on experimenting with the periods' duration in an effort to optimize model performance.

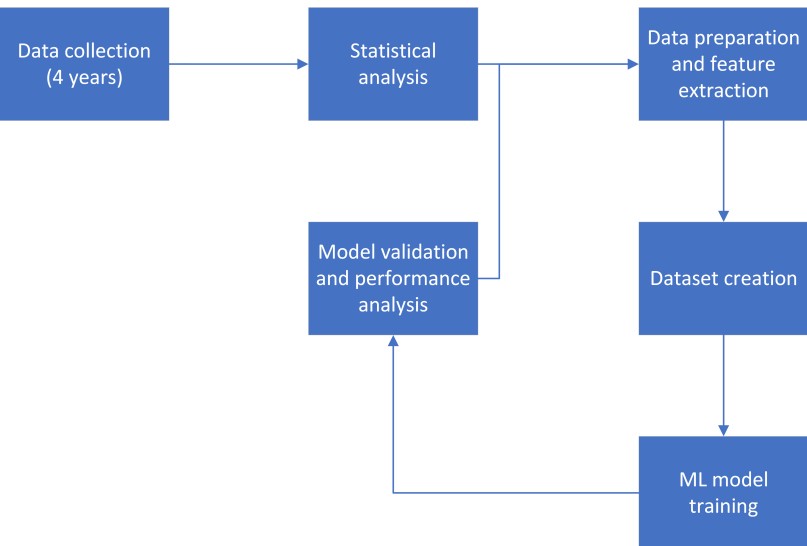

**Figure 6.** Research methodology flowchart.

### 4.1. Dataset collection

The research was conducted on a dataset provided by an internationally established game developer and consists of over 4 years' worth of player event logs from January 2018 to January 2022. The game tracks in-game player activity and creates logs which are then processed and imported into a database, as shown in Figure 7. Each log in the database represents one player-caused event that has taken place in the game and was subsequently recorded by the games' system. Based on the event invoked by the player, logs have a different label. For example, when the player starts the game, a *GameStart* event is logged, while quitting the game triggers a *GameEnd* event. Another example is when a player finishes reading through a story, the *StoryCompleted* event is triggered and logged. Each event consists of various common attributes such as the device ID (which is the primary key in the database), players' usernames (which were anonymized), the time at which the event was logged, and others. Events also contain event-specific attributes. For example, the *StoryCompleted* event will have the story ID, or the *ChoiceMade* event will contain the ID of the choice that was selected. As for the database context, each event is represented as a

separate relation table with event attributes matching the table attributes in the database. The data from the database is then subsequently processed and finally structured into a dataset fit for the selected machine learning models.

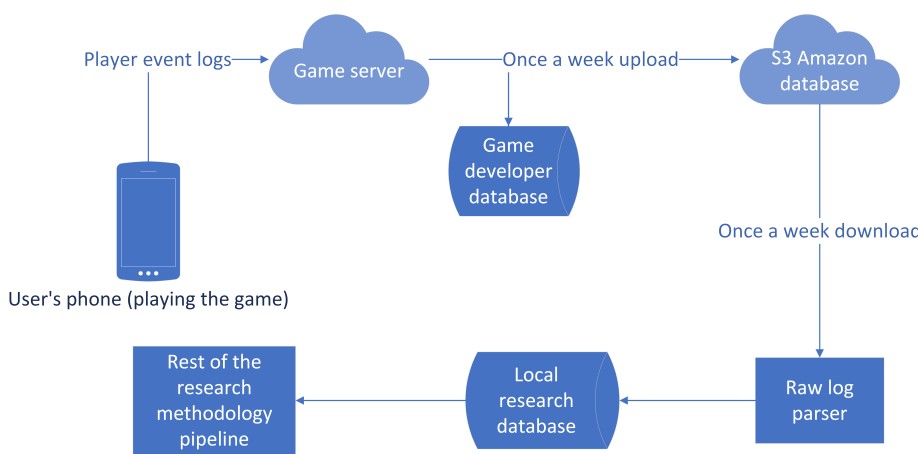

**Figure 7.** Data pipeline flowchart.

Because of the sheer number of concurrent users, and the speed at which the logs are created, we briefly outline certain errors observed in the collected data, as such input may be of interest to other researchers collecting similar large datasets with large amounts of parallel data being collected. While most observed errors are non-essential to the current research topic, there are some that severely hinder the research validity. For example, one user's session is defined as a set of events that occurred between a start and an end point (start and end events). However, in certain cases, we observed that either the start or the end event from a session was not present in the collected data. That leads to unknown start or end times for a session and an inaccuracy in the calculations relevant to user sessions (such as session duration, average session duration, etc.). Even if all the essential events for reconstructing a specific user session are present, certain events may be missing their session number and, as such, cannot be attributed to a certain session. Finally, sessions with an end-event report happening before the start event are also present in the data. Because of this, additional data cleaning is needed before the final datasets can be reliably constructed. The percentage of users removed by these methods varies on the current task, but generally, no fewer than 73.08% were removed from the initial dataset.

*4.2. Data Preparation*

Data stored in event logs must be adequately prepared for use as input for machine learning algorithms. This form is called a user profile. The dataset, however, is riddled with several imperfections, as mentioned earlier, which must be sanitized by introducing a preprocessing step before feeding the data to an algorithm. All such preparations were completed using scripts written in Python.

The dataset was filtered using first install time parameter values, and only players which started playing within the observed period were taken into consideration. Users whose first install time was wrongly recorded (or not recorded at all) were removed. Players with events that preceded their noted first install time were not included in the analysis either.

By far, the most impactful issue was the inconsistent reporting of player session start and end events. Session end events are commonly misreported, either being logged as having occurred after the start event of the next session, or simply missing altogether. This is related to players usually not quitting the game through the menu, but just switching to another application or locking the phone. Similarly (but far less frequently), sessions begin without an appropriate start event. As predicting player churn relies heavily on parameters calculated on a per-session basis (such as session length, number of certain

events in a session, etc.) this flaw represents a considerable problem which must be somehow alleviated without skewing the data.

The approach deemed most appropriate was simulating the missing events by sampling time differences from correctly reported session start and end events. A cutoff time value of 5 min was experimentally determined and further validated by the game developer. This cutoff value represents time (in seconds) of inactivity after which an ongoing session is marked as ended without a proper session end event. Sessions that commence with an event other than session start are, correspondingly, classified as incorrectly started. Time differences between accurately recorded game start logs and the first following event are sampled into a file. Likewise, this is conducted with the times between session endings and the events preceding them. When calculating features of a user profile, missing start events are compensated by inserting a faux-event at a sampled number of seconds before the first logged event of the session. The sample is chosen at random to preserve the probability distribution, with the added condition that it places the start event after the previous session's end. A similar procedure is used to compensate for absent end events—end time is calculated as a random sample, added to the time of the last recorded event. The resulting timestamp must be less than the start time of the next session.

Other notable issues include the unreliability of the lifetime value parameter (total money the user spent on the game up until the point of logging) and considerable flaws in session number markers, which were often recorded as negative values. As a result, the session number parameter is entirely ignored. Lifetime value is recalculated, a process made significantly easier by including only players that first installed the game in the observed period of 2 years. Recalculation was performed by setting the initial value of the parameter to 0 and adding to it every time a relevant event is processed (buying of in-game funds, additional content, etc.).

Corrected data was then used to compute features which constitute user profiles. User profiles are formalized representations of player activity. Each profile represents a single user and consists of the same features, parameters presumed relevant for resolving the problem at hand. For predicting player churn, those parameters are, for example, average session duration, money spent, and completed chapter count during a certain period (the first three days of play, for instance). The features' values hold significance in the designation of a player as a churner or a non-churner and quality feature selection is arguably the most important step of the process, as features of low relevance often steer the algorithm in the wrong direction, resulting in suboptimal performance. User profiles, once assembled, are suitable inputs for the machine learning algorithms.

*4.3. Datasets*

The datasets used in the statistical analysis and the two subsequent research parts of the paper had to be tailor-made to fit the specific theme of the current research topic. An overview of the datasets used in the research is shown in Table 1. The statistical analysis (reported in Section 3.2, Initial Churn Analysis) focused on player in-game choices; thus, the corresponding dataset was comprised mostly of story- and chapter-related events (such as which story was unlocked, which choice was made, which chapter was completed, etc.). The dataset contained 4 weeks' worth of data (26 October until 22 November 2020) and consisted of 819,643 unique players, thus providing insights into the underlying statistical trends. This dataset is referred to as D1.

The first part of the research focused primarily on player behavior, and as such, the dataset used focused mostly on player sessions, time spent in each session, money spent each day, and an average look at the number of events triggered by the player. The dataset contains the user profile data from 243,005 users collected over a period of four months, starting from 27 January 2018. This dataset is further referred to as D2.

The final part of the research focuses on a more in-depth look at player behavior. The dataset expanded on the original ideas from the first two parts and combined monitoring player sessions as well as player choices. Additionally, player engagement was

monitored by tracking the event distribution throughout the given time period. The dataset contains the user profile data from 427,138 users collected over a period of one month, starting from 27 April 2020. This dataset is further referred to as D3.

The datasets were sampled from different points in the game's history so as to showcase that the problem persists through different points in a game's lifecycle.

**Table 1.** Dataset D1, D2, and D3 details.

| Dataset | Sampling Period | Size (Amount of Users) | Use Case |
|---------|-----------------|------------------------|----------|
| D1 | 26 October 2020–22 November 2020 | 819,643 | Initial churn analysis focusing on player in-game choices. |
| D2 | 27 January 2018–27 May2018 | 243,003 | First part of the research focusing on player behavior. |
| D3 | 27 April 2020–27 May 2020 | 427,138 | Final part of the research focusing on a more in-depth look at player behavior. |

*4.4. Metrics for Model Evaluation*

Determining which of the selected machine learning models (previously fitted with the appropriate data) yields the best performance on a given task was accomplished by comparing the selected models using different evaluation metrics. Metrics used for model evaluation include the following: F1-score, area under the ROC curve (AUC curve), and the AP curve. The F1 score of a model is the harmonic mean of the models' raw accuracy and recall. It is considered a more reliable metric for model evaluation then its raw components, but further comparisons using AUC and AP metrics are also useful. The AUC metric is defined as the area under the ROC curve. The ROC curve is a graphical representation of the recall value of a model in relation to the FPR. Drawing the ROC curve and calculating the area underneath, we obtain the AUC metric. The AUC provides an aggregate measure of performance across all possible classification thresholds. The larger the area the ROC curve covers, the higher the model performance. Similarly to the AUC metric, the AP metric is the area covered by the precision–recall curve. Again, the bigger the area, the higher the model's performance.

## 5. Results

*5.1. Initial Churn Prediction*

The first machine learning model was trained with a feature extraction period of 10 days after installation. Using features calculated from the first 10 days of play, the model attempted to predict whether the user would churn within the following 7 days (days 11 to 17). The goal was to evaluate the initial assumptions for gameplay and relevant features for prediction on "stable" users which survived the first 10 days. For this approach, user profiles consisted of the following features: session count, average session duration, average event count per session, and average money spent per day, with each of the parameters being calculated for the first 1, 3, 7, and 10 days of play, totaling 16 features.

The values were adjusted using Scikit-learn's MinMaxScaler class on an interval of (0, 1) and fed to the RF algorithm. This model was experimentally determined to yield the best results when compared with models utilizing other available scaling solutions in combination with popular classification algorithms (linear regression, naive Bayes, K-nearest neighbors, decision tree, and linear discriminant analysis), using 30-fold cross-validation. The model's performance was deemed subpar, as it only achieved a classification precision of 56%, which, however, is high enough to warrant additional optimization. The initial model was only a proof of concept on a large dataset. It also indicated which features should be redefined—features showcasing player activity during the whole observed period, as well as further gameplay-specific features. In addition, the feedback from the game developer was to focus on shorter time periods to include the players who are active in the

game over a shorter time period. Recognizing those players early would enable possible actions which would increase the chances of retaining those players.

### 5.2. Churn Prediction for Varying Periods

Further research was conducted using variable period duration and an adjusted feature set, as well as additional metrics. Average session count, duration, and event count were retained from the first attempt, while average money spent per session was replaced with the lifetime value parameter. Several new features were introduced: chapter start, completion and drop counts, story unlock, and completion counts, as well as level of engagement and the engagement vector. While the count features are straightforward, the final two require further explanation.

Level of engagement is a feature that models playtime unevenness throughout the feature extraction period. It is calculated as the difference in event count between the first and last 30% of the period. A negative value indicates that the user played more toward the end of the observed time interval and is thus less likely to churn, as opposed to a player whose level of engagement is positive.

An engagement vector offers a more detailed view of player activity distribution. It consists of 10 elements, each representing 10% of the feature extraction period's total running time. An element's value is the percentage of events in the entire period that were logged in the corresponding interval. For example, the third element's value is equal to 0.25 if 25% of recorded events happened between 20% and 30% of the feature extraction period. These new features proved highly valuable for increasing the performance of obtained models.

Various combinations of feature extraction and churn period duration were extensively tested, coupled with several machine learning algorithms. For feature extraction, intervals of 1, 2, 3, and 7 days were considered, while churn was predicted over a period lasting between 1 and 7 days, for a total of 28 combinations. The resulting profiles were used as input for logistic regression, random forest, and K-nearest neighbors.

Models constructed in this manner were evaluated using previously described metrics. Judging solely based on precision, the RF algorithm's 72% would prevail, while K-nearest neighbors ranks the lowest with 69%. However, in terms of F1 score, logistic regression achieved the highest result of 0.78.

Taking AUC (Figure 8) and AP (Figure 9) into account, the RF algorithm exhibits the best performance for the given feature set, as shown in Table 2. It should be noted that the listed figures present the performance of models in which the feature extraction period is one day, while churn period is also one day. When compared to initial results, the accuracy is significantly higher, which can be attributed to new features. As expected, the accuracy and recall increase with longer periods of feature extraction, and decrease slightly the longer the churn period is (as certain people will not show on the first day and will be labeled as churners, but will show on the second day, as showcased by Figure 2). Overall, the accuracy of the model prediction varies from 66% to 95%.

**Table 2.** Random forest classifier results—(accuracy, recall) pairs.

| Churn Period | Feature Extraction Period | | | |
|---|---|---|---|---|
| | **1 Day** | **2 Days** | **3 Days** | **7 Days** |
| 1 day | 0.72, 0.73 | 0.87, 0.89 | 0.91, 0.92 | 0.95, 0.96 |
| 2 days | 0.69, 0.69 | 0.85, 0.87 | 0.89, 0.91 | 0.94, 0.96 |
| 3 days | 0.68, 0.67 | 0.83, 0.85 | 0.88, 0.90 | 0.93, 0.95 |
| 4 days | 0.67, 0.66 | 0.83, 0.85 | 0.87, 0.89 | 0.92, 0.95 |
| 5 days | 0.66, 0.66 | 0.82, 0.84 | 0.86, 0.88 | 0.92, 0.94 |
| 6 days | 0.66, 0.65 | 0.81, 0.83 | 0.85, 0.88 | 0.91, 0.94 |
| 7 days | 0.66, 0.65 | 0.81, 0.83 | 0.85, 0.87 | 0.91, 0.93 |

The ROC curve for this model is shown in Figure 8 and is used to calculate the AUC metric.

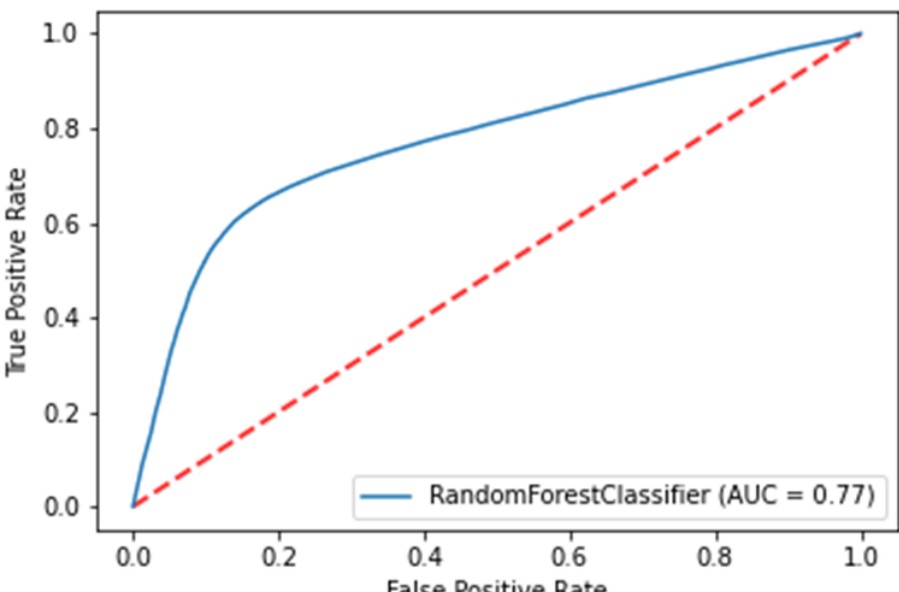

**Figure 8.** ROC curve for random forest (corresponds to models in which the feature extraction period is one day, while churn period is also one day).

The AP (average precision) metric is computed as average precision over recall. The precision–recall curve is presented by Figure 9.

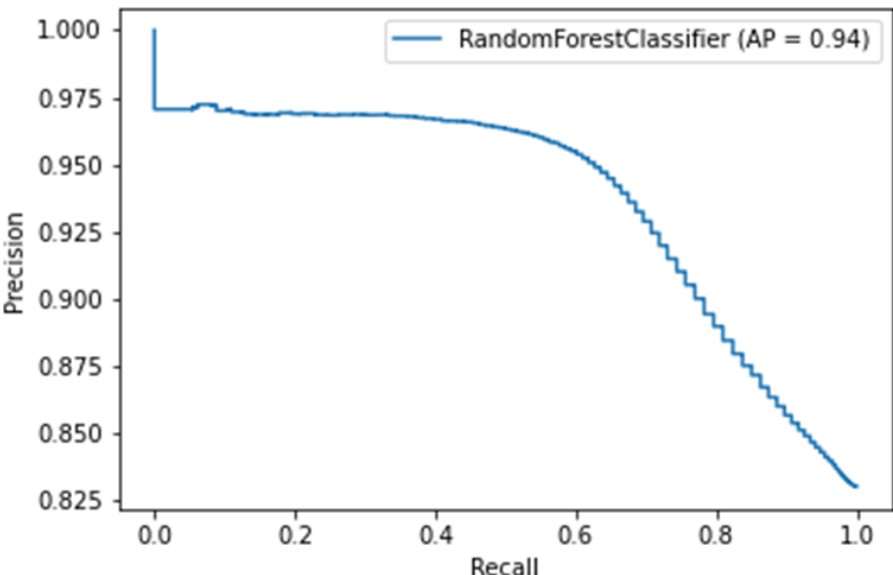

**Figure 9.** Precision over recall for random forest (corresponds to models in which the feature extraction period is one day, while churn period is also one day).

Additionally, precision, recall, and F1 score by feature extraction period duration are presented in Figure 10, with the churn period being 1 day, while the same curves calculated by changing the churn period length can be seen in Figure 11, with the feature extraction period corresponding to one day. It is interesting to note that, based on an only 1-day (feature extraction period), the precision, recall, and F1 metrics can be calculated for a 7-day churn period with values around 65%. Judging by raw data, this seems to be a worse result than the results when setting the churn period to only one day (values around 72%),

but when looking at the accuracy of the churn label estimate (as shown in Figure 3) on which the model was trained, the model is trained on more accurate labels and, thus, is expected to perform better on other, never before seen datasets. It should also be noted that new estimations can be made for each passing day. Estimation accuracy based on a 7-day learning period for the next day is above 95%, and for the next seven days is above 90%. Moreover, these values are very competitive when comparing to the results in the related work (e.g., [15]).

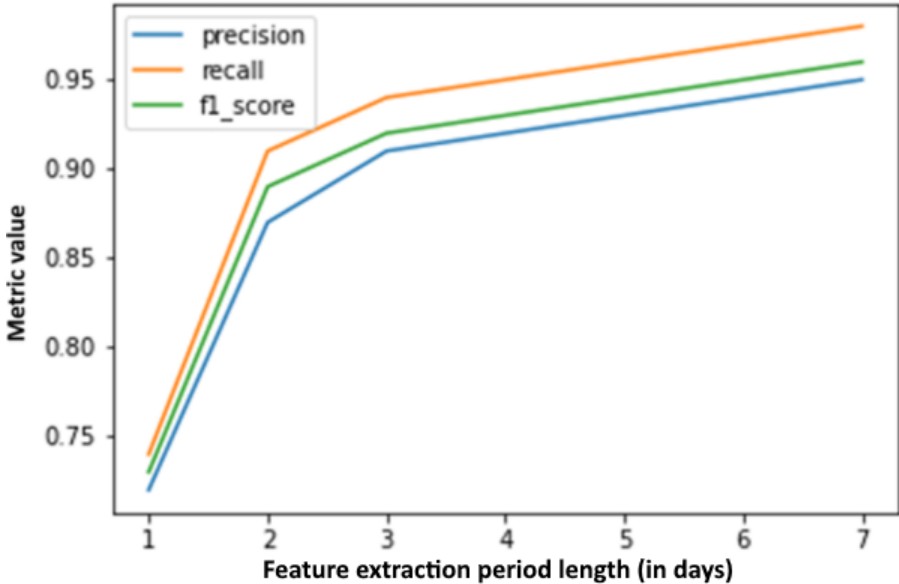

**Figure 10.** Metric changes by feature extraction period duration.

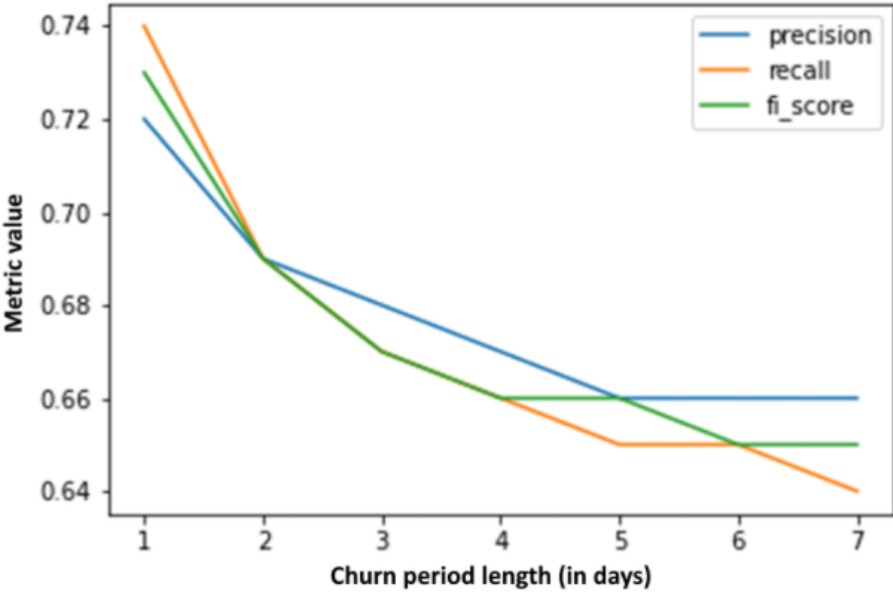

**Figure 11.** Metric changes by churn period duration.

With a classification precision of 95% and a 0.96 F1 score, optimal durations are clearly 7 days of feature extraction for 1 day of churn prediction. The confusion matrix for this model is given in Figure 12.

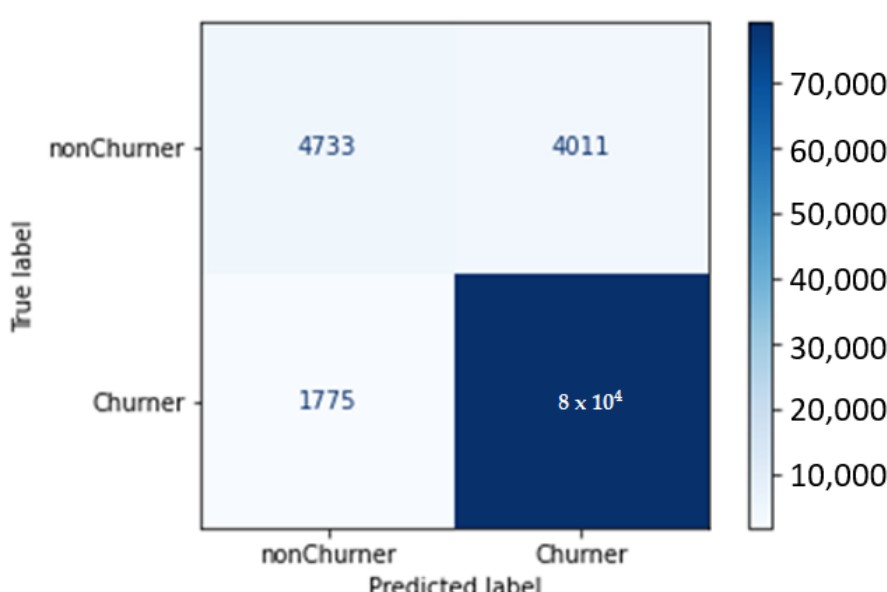

**Figure 12.** Confusion matrix—7 days of feature extraction and 1 day of churn prediction.

However, as a disproportionately high percentage of players churn after a single day of use, observing 7 days of a player's actions is not universally applicable; thus, a far more practical combination would be 1 day of feature extraction and 1 day of prediction. In other words, calculating features based on the first day of each user's playtime and predicting whether they will churn the next day fits our use case better. Since that particular length combination achieved 72% precision and a 0.73 F1 score (confusion matrix presented in Figure 13), numbers hopefully high enough to warrant use in production, the research is considered successful with enough space for further optimization. Based on feedback from the developer, there is even interest in models which try to predict the churn based on only several hours of gameplay.

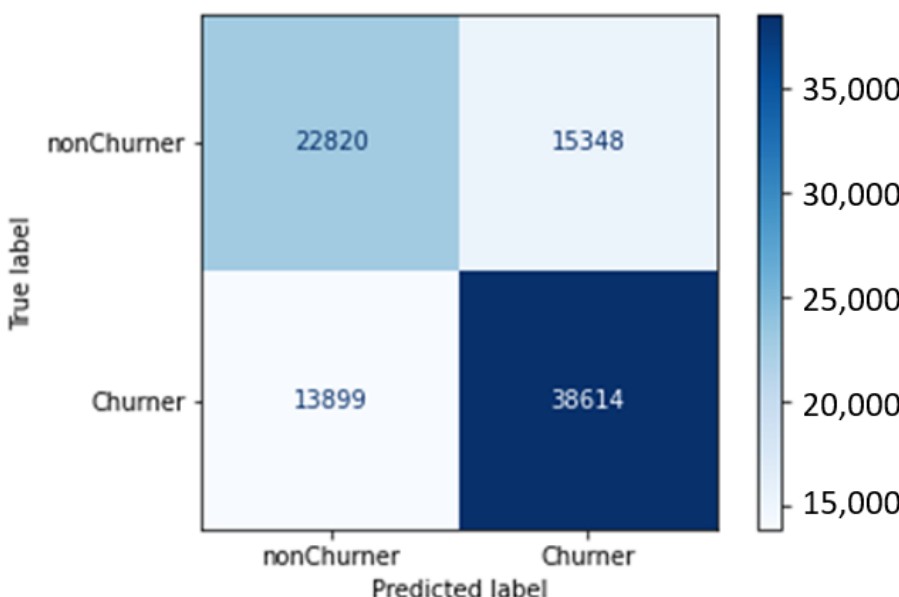

**Figure 13.** Confusion matrix—1 day of feature extraction and 1 day of churn prediction.

## 6. Conclusions and Outlook

This paper presents our efforts to apply machine learning techniques to predict player churn of a globally popular mobile F2P storytelling game. Following the collection of a very large dataset over the course of a 4-year period, we discuss the process of data collection and cleaning in preparation for training various ML models. Our initial statistical analysis of the data shows significant user churn within 1–7 days of initial game installment, and thus motivates the need for developing ML-based models to accurately predict such events. This challenge is even more pronounced when considering our specific studied game as compared to related work, due to the very dynamic player behavior in the studied game. Timely reactions can prevent the potential departure of players, for example, by providing special offers, additional tickets, etc. The statistical review of the data also justifies the use of simple end-of-interest assumptions as valid tags for user feature vectors (churnLabel tags), as the accuracy of the simple assumption ranges from 83.65% to more, as interest-end observation days increase. Furthermore, we notice the problems in the implementation of tracking a large number of players concurrently, and when determining how this can affect the validity of logged information. We perform empirically based data preparation to alleviate these issues.

The most important implications of the study are as follows. The ML models can predict, with significant accuracy, the departure of players from the game even at very short learning and prediction periods (i.e., one day). The data analysis suggests that, for this game, and possibly for this game genre, even shorter time frames need to be considered (i.e., hours). The prediction for shorter time frames will enable developers to react quickly to players potentially leaving the game through specific offers or rewards in an effort to try to keep the players engaged. We find the use of features which specify the distribution of player activity during the observed time period to be of the highest significance for the performance of ML models. Thus, we encourage future research to encompass such features in their models.

From the tested ML algorithms, the random forest model stood out as the best model in the general analysis, while the logistic regression resulted with the poorest performance. Considering a period of one day of user observation and one day of churn period AUC, metrics for the algorithm of random noise of 0.77 and average AP accuracy of 94% are good results for short time periods of observation, and the reliability of this algorithm is satisfactory. Nevertheless, when considering the presented results, the behaviors of the players and behavior variations over time need to be taken into account. To illustrate, in the first day, over 60% of the players were found to become churners, while, when considering periods after the 10th day, only 5% of the remaining players will churn.

The results of this study should be considered with several limitations. First, the study was conducted on a casual interactive storytelling game. Further study is needed to confirm whether similar behavior is observed in other games of this genre and whether similar methodologies and models can be applied within this game's genre and other game genres. Additionally, the event-logging system of the game was flawed, so the measures used for the reconstruction of the players' sessions may have impacted the data. Another limitation of the research was in the computational power required for processing such large datasets, so the initial dataset consisting of 4 years of data needed to be broken down into smaller ones for faster and more convenient ML model iterations. The study was conducted during a period of the game's decline, so that specific point in the lifecycle of the game also needs to be taken into account.

Our ongoing and future research in this area will focus on considering even shorter time periods for prediction periods shorter than one day (i.e., hours following game installment). This direction was advised by the game developer as well. We also aim to investigate the applicability of deep learning methods, such as neural networks, in this regard. Furthermore, we aim to evaluate the monetary performance of players and to create models for the prediction of player purchases.

**Author Contributions:** Conceptualization, M.S.; methodology, K.B., K.M., L.S.-K. and M.S.; software, K.B. and K.M.; validation, K.B., K.M., L.S.-K. and M.S.; investigation, K.B., K.M., L.S.-K. and M.S.; resources, L.S.-K. and M.S.; writing—original draft preparation, K.B., K.M., L.S.-K. and M.S.; writing—review and editing, L.S.-K. and M.S.; visualization, K.B., K.M.; supervision, L.S.-K. and M.S.; project administration, L.S.-K. and M.S.; funding acquisition, L.S.-K. and M.S. All authors have read and agreed to the published version of the manuscript.

**Funding:** This work has been supported in part by the Croatian Science Foundation under the project IP-2019-04-9793 (Q-MERSIVE).

**Institutional Review Board Statement:** Ethical review and approval were waived for this study due to the fact that research described in this paper is conducted on an anonymous dataset which is gathered automatically from the players.

**Informed Consent Statement:** Player consent was waived due to he fact that research described in this paper is conducted on an anonymous dataset which is gathered automatically from the players. There is no way to identify a specific person in the dataset.

**Data Availability Statement:** Restrictions apply to the availability of these data. Data was obtained from the game developer in scope of a research project and is not publicly available.

**Conflicts of Interest:** The authors declare no conflict of interest.

## Abbreviations

The following abbreviations are used in this manuscript:

| | |
|---|---|
| AP | average precision |
| AUC | area under curve |
| DLC | downloadable content |
| F2P | free-to-play |
| FPR | false-positive rate |
| ML | machine learning |
| MMORPG | massively multiplayer online role-playing game |
| P2P | pay-to-play |
| RF | random forest |
| RFM-LIR | recency, frequency, monetary value–lifetime, intensity, rewards |
| ROC | receiver operating characteristic |
| WoW | *World of Warcraft* |

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
