# Peer review of "Predicting Player Churn of a Free-to-Play Mobile Video Game Using Supervised Machine Learning"

_applsci, doi:10.3390/app12062795_

Round 1
Reviewer 1 Report
Thank you for providing me with the opportunity to read “Predicting player churn of a free-to-play mobile videogame using supervised machine learning”. I have the following comments:
- The paper lacks support from relevant literature. This is clearly evident and supported by the lack of references. The authors need to extensively add to the related literature section and support their study using state-of-the-art literature.
- I am a little skeptical about the keyword “player behavior analysis” I suggest removing it or revising the paper to incorporate behavior analysis in actuality.
- Provide a table/list of abbreviations in the paper (can be at the start or an appendix)
- The novelty of the study is not clear. This is a serious shortcoming and must be properly addressed to warrant publication.
- The introduction doesn’t justify why we need to use ML model for this purpose. This needs to be clearly explained in a detailed paragraph
- More details about the case study game are needed. What makes it a useful case study must be justified and elaborated on in detail.
- What behavioral parameters were used in this study? This needs to be clarified and supported by relevant literature as to why and how these are the best fit for representing player behavior in online environments/games.
- Please clearly state the limitation of the study in the conclusions.
- The implications of this study for research and practice need to be clarified.
Author Response
Dear reviewer, thank you for your valuable reviews which have significantly improved our paper. Our responses are included in the attached file.

Reviewer 2 Report
- Lines 83-91, it is unclear how the original dataset was split to construct smaller datasets;
- Lines 187-191, If the user first installed the game on a certain date and the very next day the system has not logged any activity from the same user, then the user is estimated to have churned and the churn label would be positive. Alternatively, if no activity is detected on the next day following installment, the churn label would be set to negative – the player has not churned. (check please for a misprint, if not, construction seems to be unclear and overcomplicated);
- Figure 7 contains too small elements and looks uncomfortable to watch;
- Lines 271-285, is it possible to bring some percentage values of removed dataset elements for the described cases?
- Line 434, extra “(“ ?;
- Lines 444-446, can you give some confirmation, if it can be recognized as a good result?
- It remains unclear to me if the study is representative of other game genres or does each require its own study?
Author Response

(The authors gave the same response as above.)

Round 2
Reviewer 1 Report
I am happy with the changes made. Thank you and all the best.